# Biphasic regulation of chondrocytes by Rela through induction of anti-apoptotic and catabolic target genes

Hiroshi Kobayashi[1], Song Ho Chang[1], Daisuke Mori[1,2], Shozo Itoh[1], Makoto Hirata[1], Yoko Hosaka[1,2], Yuki Taniguchi[1], Keita Okada[1], Yoshifumi Mori[1], Fumiko Yano[2,3], Ung-il Chung[3], Haruhiko Akiyama[4], Hiroshi Kawaguchi[1,5], Sakae Tanaka[1] & Taku Saito[1,2]

*In vitro* studies have shown that Rela/p65, a key subunit mediating NF-κB signalling, is involved in chondrogenic differentiation, cell survival and catabolic enzyme production. Here, we analyse *in vivo* functions of Rela in embryonic limbs and adult articular cartilage, and find that Rela protects chondrocytes from apoptosis through induction of anti-apoptotic genes including *Pik3r1*. During skeletal development, homozygous knockout of *Rela* leads to impaired growth through enhanced chondrocyte apoptosis, whereas heterozygous knockout of *Rela* does not alter growth. In articular cartilage, homozygous knockout of *Rela* at 7 weeks leads to marked acceleration of osteoarthritis through enhanced chondrocyte apoptosis, whereas heterozygous knockout of *Rela* results in suppression of osteoarthritis development through inhibition of catabolic gene expression. Haploinsufficiency or a low dose of an IKK inhibitor suppresses catabolic gene expression, but does not alter anti-apoptotic gene expression. The biphasic regulation of chondrocytes by Rela contributes to understanding the pathophysiology of osteoarthritis.

[1] Department of Sensory & Motor System Medicine, Faculty of Medicine, The University of Tokyo, 7-3-1 Hongo, Bunkyo-ku, Tokyo 113-8655, Japan.
[2] Department of Bone and Cartilage Regenerative Medicine, Faculty of Medicine, The University of Tokyo, 7-3-1 Hongo, Bunkyo-ku, Tokyo 113-8655, Japan.
[3] Division of Clinical Biotechnology, Center for Disease Biology and Integrative Medicine, Faculty of Medicine, The University of Tokyo, 7-3-1 Hongo, Bunkyo-ku, Tokyo 113-8655, Japan. [4] Department of Orthopaedic Surgery, Gifu University, 1-1 Yanagito, Gifu 501-1193, Japan. [5] Department of Spine Center, Tokyo Shinjuku Medical Center, Japan Community Health Care Organization, 5-1 Tsukudotyo, Shinjuku-ku, Tokyo 162-8543, Japan. Correspondence and requests for materials should be addressed to T.S. (email: tasaitou-tky@umin.ac.jp).

Chondrocyte differentiation is an essential process for endochondral ossification[1]. During skeletal development, mesenchymal progenitor cells are recruited into condensations and differentiate into chondrocytes that produce cartilage-specific matrix proteins such as type II collagen (Col2a1) and aggrecan (Acan). This process is regulated by the sex-determining region Y-type high-mobility group box protein (Sox9). The cartilage enlarges through chondrocyte proliferation and matrix production. Later, chondrocytes cease proliferation and undergo hypertrophic differentiation characterized by secretion of type X collagen (Col10a1). Finally, hypertrophic chondrocytes undergo apoptotic cell death, and the cartilage matrix is degraded for the proceeding bone formation[1].

In adulthood, chondrocytes maintain articular cartilage homeostasis, and its disruption results in osteoarthritis (OA), the most prevalent joint disorder with articular cartilage degradation[2,3]. A disintegrin-like and metallopeptidase with thrombospondin type 1 motif 5 (Adamts5), a representative protease of Acan, has been shown to be involved in OA development using a mouse OA model[4,5]. Previously, we showed that hypoxia-inducible factor 2 alpha (HIF2a) tightly regulates cartilage degradation through transcriptional induction of various catabolic genes[6,7]. Expression of HIF2a in articular cartilage increases during the early and middle stages of OA, and we further identified nuclear factor kappa B (NF-κB) signalling as a direct transcriptional inducer of HIF2a in OA development[6].

The NF-κB family of transcription factors has essential roles in a wide range of biological processes such as immune responses, inflammation, proliferation, differentiation, cell survival and apoptosis[8–10]. This family includes v-rel reticuloendotheliosis viral oncogene homologue A (Rela, also known as p65), Relb, Rel, p105/p50 and p100/p52, each of which includes a Rel homology domain that mediates DNA binding and dimerization. These proteins usually act as transcription factors after hetero-dimerization. Inhibitors of NF-κB (IκB) proteins, which have several members, including as IκBα, IκBβ, IκBγ, IκBε, IκBζ and Bcl-3, bind to some NF-κB family members in the cytoplasm[11]. Upon activation of IκB kinases (IKKs) in response to several signals, IκB proteins are phosphorylated and degraded. Different IKK complexes function together to mediate canonical and non-canonical NF-κB signalling, mainly via IKKβ and IKKα, respectively. The degradation of IκB proteins enables free NF-κB complexes to translocate from the cytoplasm into the nucleus, leading to target gene transactivation[12,13]. NF-κB family genes are expressed in chick limb cartilage, and blockade of NF-κB signalling or deletion of IKKα causes limb outgrowth impairment[14–16]. In addition, Rela supports chondrocyte survival by activating Nkx3.2 (ref. 17). We previously reported that Rela is a potent transcription factor of Sox9 (ref. 18), which is involved in the regulation of chondrocyte differentiation during skeletal development via glycogen synthase kinase-3α and glycogen synthase kinase-3β (ref. 19). Furthermore, we revealed that Adamts5 is a direct transcriptional target of Rela in chondrocytes[20]. Hence, Rela may be extensively involved in anabolic and catabolic processes of cartilage during skeletal growth, articular cartilage homeostasis and OA development.

Here, we examined the *in vivo* roles of Rela in embryonic limb cartilage and adult articular cartilage using various tissue-specific knockout mice. During skeletal development, homozygous knockout of *Rela* resulted in impaired growth through enhanced chondrocyte apoptosis, whereas heterozygous knockout of *Rela* did not alter growth. In articular cartilage, homozygous knockout of *Rela* led to marked acceleration of OA through enhanced chondrocyte apoptosis, whereas heterozygous knockout of *Rela* resulted in suppression of OA development through inhibition of catabolic gene expression. Our findings provide new insights into the regulation of articular cartilage homeostasis and OA development.

## Results

**Expression of Rela and IκB in growth plate chondrocytes.** We first examined the expression and localization of Rela and IκB proteins in the epiphyseal cartilage of mouse embryos by immunofluorescence. Rela protein was localized in both the cytoplasm and nucleus throughout the growth plate, indicating no obvious subcellular translocation of Rela protein (Fig. 1a,b). IκB protein was also detected in periarticular and hypertrophic chondrocytes, while phosphorylated IkB was weakly detected in periarticular chondrocytes and decreased in hypertrophic chondrocytes (Fig. 1c,d). These data indicate that the canonical NF-κB pathway is not highly activated during chondrocyte differentiation.

**Role of Rela in chondrocytes during skeletal development.** To examine the physiological role of Rela in skeletal development, we deleted Rela in undifferentiated limb mesenchyme, whole chondrocytes or hypertrophic chondrocytes by mating a paired-related homeobox gene 1 (Prrx1) promoter-driven Cre-transgenic mouse (Prrx1-Cre)[21], Col2a1 promoter-driven Cre-transgenic mouse (Col2a1-Cre)[22] or Col10a1-Cre knock-in mouse (Col10a1-Cre)[23] with a mouse homozygous for a floxed Rela allele (Rela^{fl/fl})[24] (Supplementary Fig. 1a). Recombination of the floxed allele in Prrx1-Cre; Rela^{fl/fl} and Col2a1-Cre; Rela^{fl/fl} chondrocytes was confirmed by PCR (Supplementary Figs 1b and 2a). Tissue-specific deletion of *Rela* was confirmed by PCR with reverse transcription (RT–PCR) using cDNA samples of various tissues obtained from Prrx1-Cre; Rela^{fl/fl} and Col2a1-Cre; Rela^{fl/fl} mice (Supplementary Figs 1c and 2b). Prrx1-Cre; Rela^{fl/fl} and Col2a1-Cre; Rela^{fl/fl} mice exhibited significantly shorter limbs than their respective control (Rela^{fl/fl}) littermates during the perinatal period (Fig. 2a–g). Axial length, vertebral length and body weight were decreased in Col2a1-Cre; Rela^{fl/fl} mice but unaffected in Prrx1-Cre; Rela^{fl/fl} mice, probably because Prrx1 is not expressed in vertebra[21] (Fig. 2c,g). Despite the growth impairment, the proportion of each zone was not obviously altered between these knockout mice and their control littermates (Fig. 2d,h). Col10a1-Cre; Rela^{fl/fl} mice exhibited normal skeletal growth, although Rela expression was efficiently decreased in the hypertrophic zone (Supplementary Fig. 3a–d), indicating that Rela is not essential for the hypertrophic differentiation of chondrocytes.

**Enhancement of chondrocyte apoptosis by Rela deficiency.** We next examined the mechanism underlying the impaired skeletal growth of tissue-specific Rela knockout mice. Although Rela was efficiently knockdown in cartilage, matrix production was unchanged by Rela knockout (Fig. 3a). Cell proliferation determined by BrdU uptake and a cell counting assay, and hypertrophic differentiation determined by immunostaining of Col10a1 were also unaffected in Rela conditional knockout mice (Fig. 3a–c). The messenger RNA (mRNA) levels of cell cycle markers, including *Cdkn1c* (also known as *p57*) and *Ccnd1* (also known as *Cyclin D1*), were also unchanged, as well as those of chondrocyte markers (Fig. 3e). However, TUNEL staining revealed significant enhancement of chondrocyte apoptosis in the proliferative and pre-hypertrophic zones of Col2a1-Cre; Rela^{fl/fl} limb cartilage (Fig. 3f,g). Furthermore, the number of cells in the proliferative zone was decreased in Col2a1-Cre; Rela^{fl/fl} mice (Fig. 3h,i). These data indicate that the decreased number of chondrocytes in Rela-deficient limb cartilage caused by ectopic apoptosis may contribute to the impaired skeletal growth.

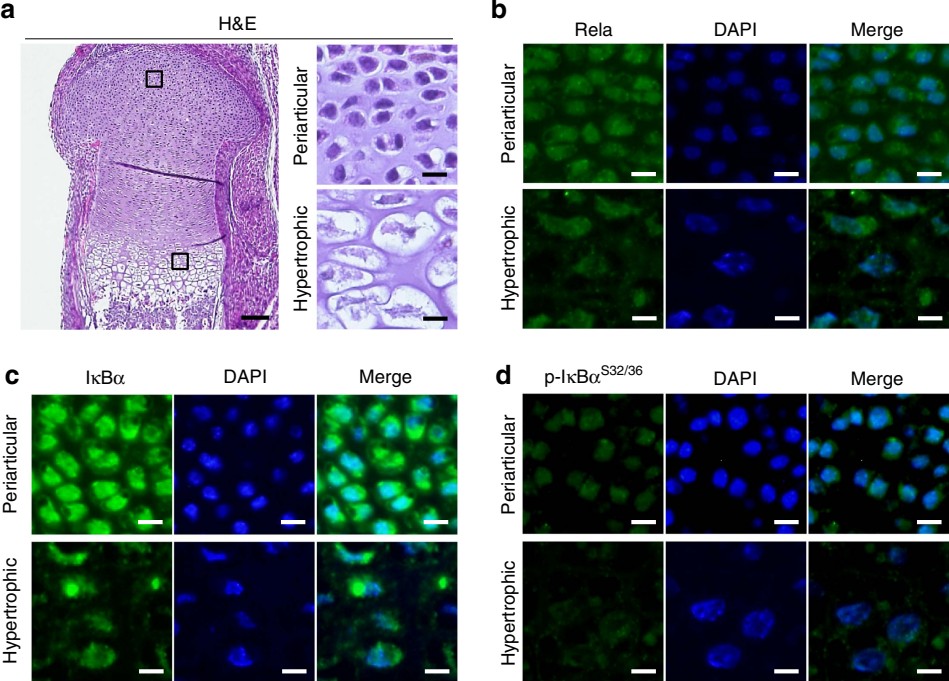

**Figure 1 | Expression of Rela and IκBα in developing chondrocytes.** (**a**) haematoxylin and eosin (H&E) staining of the epiphyseal cartilage of a mouse embryo (embryonic day (**e**) 18.5). Boxes in the left panel indicate enlarged images of the periarticular and hypertrophic zones. Scale bars, 100 and 10 μm for low and high magnification images, respectively. Immunofluorescence of Rela (**b**), IκBα (**c**), and Ser 32 and 36 dual phosphorylated IκBα (**d**) proteins in the periarticular and the hypertrophic zones. The images are representative of two independent experiments. Scale bars, 10 μm.

**Regulation of OA development by Rela**. We next investigated the contribution of Rela to articular cartilage homeostasis and OA development. We first examined the expression of Rela and IκB proteins in articular cartilage of mouse knee joints. Unlike in the epiphyseal cartilage of mouse embryos, Rela was translocated from the cytoplasm into the nucleus with cartilage degeneration caused by surgical induction of joint instability[25] (Fig. 4a). Phosphorylated IkB was strongly detected in the degenerated cartilage (Fig. 4a).

To reveal the role of Rela in articular cartilage, we performed *in vivo* loss-of-function analyses. Because *Col2a1-Cre; Rela^{fl/fl}* mice exhibited dwarfism, we used transgenic mice in which Cre recombinase was fused to a mutated ligand-binding domain of the human oestrogen receptor driven by the *Col2a1* promoter (*Col2a1-Cre^{ERT}*) to knockout *Rela* in adult articular cartilage after skeletal growth, so that the fusion protein was translocated into nuclei causing gene targeting by administration of the oestrogen antagonist tamoxifen[26]. We generated tamoxifen-inducible chondrocyte-specific homozygous knockout mice of Rela by mating *Col2a1-Cre^{ERT}* mice with *Rela^{fl/fl}* mice (*Col2a1-Cre^{ERT}; Rela^{fl/fl}*). Because *Col2a1-Cre^{ERT}; Rela^{fl/fl}* mice developed and grew normally (Supplementary Fig. 4), we injected tamoxifen into 7-week-old *Col2a1-Cre^{ERT}; Rela^{fl/fl}* mice and *Rela^{fl/fl}* littermates daily for 5 days, and established the surgical OA model at 2 days after the last injection[27]. Eight weeks after surgical OA induction, the cartilage degradation was significantly enhanced with increased chondrocyte apoptosis in *Col2a1-Cre^{ERT}; Rela^{fl/fl}* knee joints compared with *Rela^{fl/fl}* joints (Fig. 4b–d). When we further analysed OA development in 12-month-old *Col2a1-Cre^{ERT}; Rela^{fl/fl}* mice, OA development was markedly accelerated and chondrocyte apoptosis was enhanced in these mice (Fig. 4b–d). Interestingly, the expression of Hif2a and Adamts5 was suppressed in *Col2a1-CreERT; Rela^{fl/fl}* knee joints of both models (Fig. 4b). These findings indicate that Rela is a critical regulator of articular cartilage homeostasis, and its deficiency results in enhanced cartilage degradation caused by increased chondrocyte apoptosis, despite suppression of Adamts5.

**Identification of anti-apoptotic target genes of Rela**. To determine how Rela deficiency resulted in enhanced apoptosis, we first treated primary cultures of articular chondrocytes with tumour necrosis factor (TNF), an inducer of chondrocyte apoptosis[28–30]. To identify target genes mediating the anti-apoptotic effect of Rela, we isolated primary chondrocytes from *Prrx1-Cre; Rela^{fl/fl}* and *Rela^{fl/fl}* mice, treated them with or without 10 ng ml^{−1} TNF, and compared gene expression profiles by microarray analyses. Among 2,581 genes upregulated by TNF, 1,551 genes were downregulated by Rela ablation (Fig. 5a). Pathway analyses further identified 70 genes related to apoptosis, of which nine have anti-apoptotic effects according to previous studies (Fig. 5a). Among the nine genes, real-time RT-PCR confirmed that the expression of *Traf2*, *Birc2(c-Iap1)*, *Birc3(c-Iap2)* and *Pik3r1* was increased by TNF treatment and decreased by Rela deletion (Fig. 5b). Immunofluorescence confirmed that these four proteins were expressed in mouse articular cartilage and suppressed by Rela deletion (Fig. 5c).

Traf2 is a member of the TNF receptor associated factor (TRAF) family. Traf1 and Traf2 interact with the inhibitor-of-apoptosis protein (IAP) family that includes Birc2 and Birc3, and exert anti-apoptotic effects[31]. *Traf2*, *Birc2* and *Birc3* are induced by NF-κB signalling and mediate resistance against TNF-induced apoptosis[31]. Pik3r1 encodes a p85α regulatory protein that is a subunit of phosphatidylinositol 3-kinase (PI3K). It is required for the stabilization and localization of p110–PI3K activity to activate Akt (ref. 32). The PI3K/Akt pathway plays a wide range of roles such as proliferative and anti-apoptotic signalling, and insulin responses[33]. However, transcriptional regulation of *Pik3r1* by NF-κB signalling or Rela has been unknown. Because Pik3r1 was significantly downregulated by Rela deletion, we examined

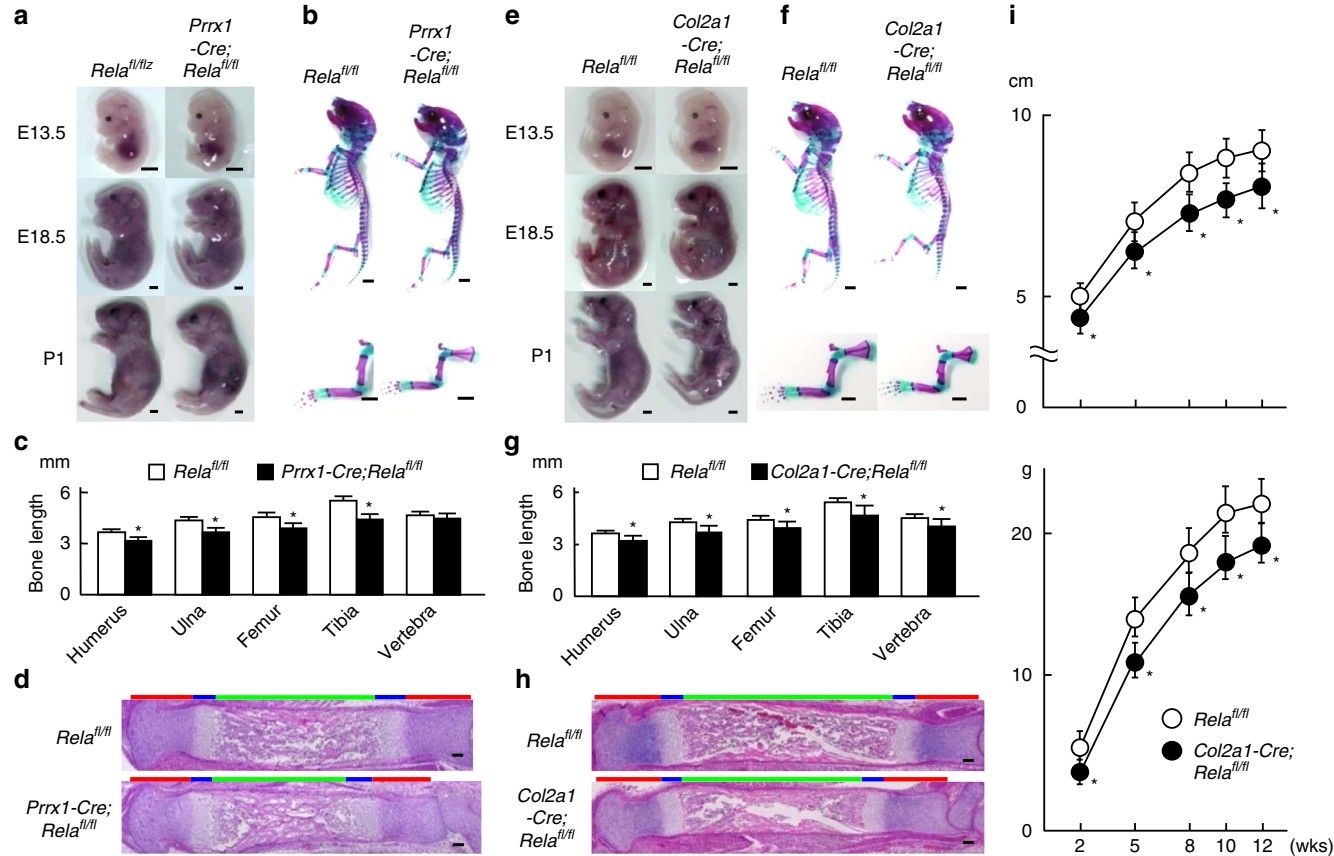

**Figure 2 | Skeletal development of *Prrx1-Cre; Rela^fl/fl* and *Col2a1-Cre; Rela^fl/fl* mice.** Gross appearance of *Rela^fl/fl* and *Prrx1-Cre; Rela^fl/fl* (**a**) or *Col2a1-Cre; Rela^fl/fl* (**e**) littermate embryos. Scale bars, 1 mm. Double staining with Alizarin red and Alcian blue of the whole skeleton (top) and upper extremities (bottom) of *Rela^fl/fl* and *Prrx1-Cre; Rela^fl/fl* (**b**) or *Col2a1-Cre; Rela^fl/fl* (**f**) littermate embryos (E18.5). Scale bars, 1 mm. Length of the long bones and vertebrae (first to fifth lumbar spines) of *Rela^fl/fl* and *Prrx1-Cre; Rela^fl/fl* (**c**) or *Col2a1-Cre; Rela^fl/fl* (**g**) littermate embryos (E18.5). Data are expressed as the mean ± s.d. of five mice per group. *$P < 0.05$ versus *Rela^fl/fl* mice; Student's *t*-test. Haematoxylin and eosin (H&E) staining of whole tibias of *Rela^fl/fl* and *Prrx1-Cre;Rela^fl/fl* (**d**) or *Col2a1-Cre; Rela^fl/fl* (**h**) littermate embryos (E18.5). Scale bars, 100 μm. Upper bars indicate lengths of the proliferative zone (red), hypertrophic zone (blue) and bone area (green). (**i**) Growth curves of *Rela^fl/fl* and *Col2a1-Cre; Rela^fl/fl* males. Height indicates the nose–anus length. Data are expressed as the means ± s.d. of five mice per group. *$P < 0.05$ versus *Rela^fl/fl* mice; Student's *t*-test.

activation of Akt by TNF in Rela knockout chondrocytes. TNF treatment enhanced phosphorylation of Akt at Thr427 in *Rela^fl/fl* chondrocytes, but not *Prrx1-Cre; Rela^fl/fl* chondrocytes, in a dose-dependent manner (Fig. 5d).

**Transcriptional induction of *Pik3r1* by Rela.** To examine the mechanism of Pik3r1 induction by NF-κB signalling, we initially performed exhaustive comparisons of sequences of ~4 kb in the 5′-end flanking regions of mouse and human genes, and found that 1.6 kb upstream from the transcription start site was conserved by ~90%. There were two NF-κB motifs in the 1.6 kb region of the human *PIK3R1* proximal promoter, N-1 (the distal motif, − 546 to − 539 bp) and N-2 (the proximal motif, − 227 to − 219 bp; Fig. 6a). In the mouse chondrocyte cell line ATDC5, RELA overexpression activated *PIK3R1* promoter activity in a dose-dependent manner of the expression vector (Fig. 6b). A series of 5′-deletions of the fragment identified decreases in the transcriptional activity between –771 and –481 bp, and between – 270 and –97 bp, containing N-1 and N-2, respectively (Fig. 6c). Hence, we next compared luciferase activities by mutagenesis, and found that the activity was significantly decreased by mutation of either motif (Fig. 6d). Chromatin immunoprecipitation (ChIP) assays further showed *in vivo* binding of RELA protein to the region including N-2 in lysates of the human chondrocyte cell line OUMS-27 stimulated by TNF, but it was not detected in the

region including N-1, indicating that N-2 is a functional motif for Rela binding (Fig. 6e).

**Suppression of OA development by haploinsufficiency of Rela.** We previously showed that activation of NF-κB signalling induces Hif2a in OA development[6], and that Rela is a transcriptional activator of ADAMTS5 in chondrocytes by *in vitro* analyses[20]. In the present study, homozygous knockout of Rela in articular chondrocytes suppressed Adamts5 expression. However, it also enhanced chondrocyte apoptosis by insufficient induction of anti-apoptotic factors, resulting in markedly accelerated OA development (Fig. 4b,c). Taken together, we hypothesized that Rela may regulate articular cartilage homeostasis in a biphasic manner. We next analysed OA development in chondrocyte-specific heterozygous knockout mice of Rela. We first confirmed that *Col2a1-Cre;Rela^fl/+* mice grew normally (Supplementary Fig. 5a–c), indicating that haploinsufficiency of Rela does not affect skeletal development. Next, we examined OA development in *Col2a1-Cre^ERT; Rela^fl/+* mice. Notably, the cartilage degradation was significantly suppressed in *Col2a1-Cre^ERT; Rela^fl/+* mice compared with control *Rela^fl/+* mice (Fig. 7a,b). Expression of Adamts5 and Hif2a was suppressed in *Col2a1-Cre^ERT; Rela^fl/+* cartilage, while chondrocyte apoptosis determined by TUNEL staining was unaltered (Fig. 7a,c). In *ex vivo* cultures of femoral head

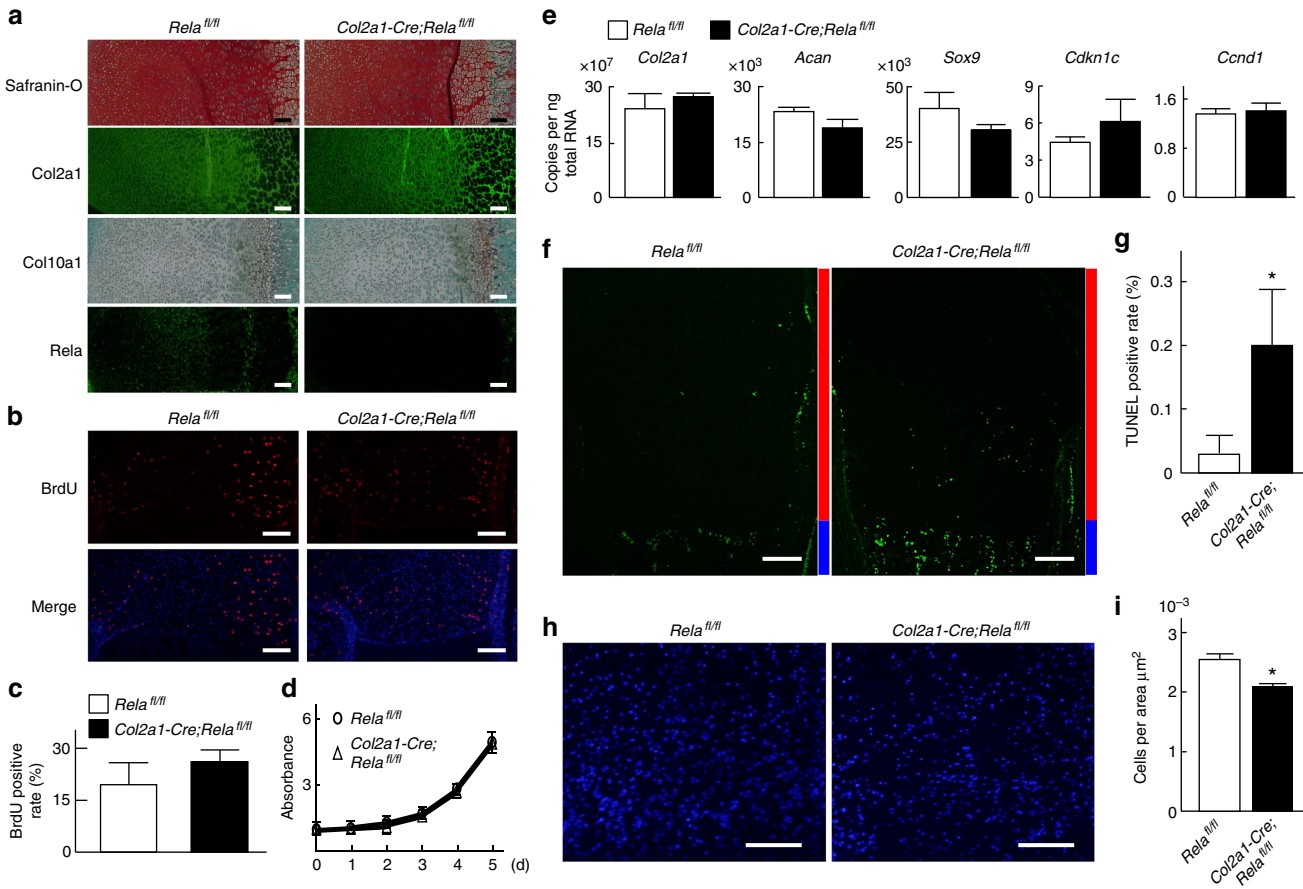

**Figure 3 | Enhanced apoptosis in epiphyseal cartilage of *Col2a1-Cre; Rela^{fl/fl}* embryos.** (**a**) Safranin-O staining and immunostaining of Col2a1, Col10a1 and Rela in proximal tibias of *Rela^{fl/fl}* and *Col2a1-Cre; Rela^{fl/fl}* littermate embryos (E18.5). The images are representative of three independent experiments. Scale bars, 100 μm. BrdU staining (**b**), and the rate of BrdU-positive cells (**c**) in the proliferative zone of proximal tibias of *Rela^{fl/fl}* and *Col2a1-Cre; Rela^{fl/fl}* littermate embryos (E18.5). The images are representative of three independent experiments. Scale bars, 100 μm. Data are expressed as means ± s.d. of three sections per group. (**d**) Cell proliferation curves determined by a CCK-8 assay during 5 days of culture of primary chondrocytes derived from *Rela^{fl/fl}* and *Col2a1-Cre; Rela^{fl/fl}* littermates. Data are expressed as the mean ± s.d. of three sections per group. (**e**) mRNA levels of *Col2a1*, *Acan*, *Sox9*, *Cdkn1c* and *Ccnd1* in primary chondrocytes derived from *Rela^{fl/fl}* and *Col2a1-Cre; Rela^{fl/fl}* littermates. Data are expressed as the means ± s.d. of three wells per group. TUNEL staining (**f**), and the rate of TUNEL-positive cells (**g**) of proximal tibias of *Rela^{fl/fl}* and *Col2a1-Cre; Rela^{fl/fl}* littermate embryos (E18.5). Scale bars, 100 μm. Data are expressed as the means ± s.d. of three sections per group. *$P < 0.05$ versus *Rela^{fl/fl}* mice; Student's *t*-test. 4,6-diamidino-2-phenylindole (DAPI) staining (**h**), and the number of DAPI-positive cells (**i**) in proximal tibias of *Rela^{fl/fl}* and *Col2a1-Cre; Rela^{fl/fl}* littermate embryos (E18.5). The images are representative of three independent experiments. Scale bars, 100 μm. Data are expressed as the means ± s.d. of three sections per group. *$P < 0.05$ versus *Rela^{fl/fl}* mice; Student's *t*-test.

cartilage from *Rela*^{fl/+} mice, the release of proteoglycans into the medium and the expression of *Adamts5* stimulated by interleukin-1β treatment was significantly decreased in *Prrx1*-Cre; *Rela*^{fl/+} cartilage compared with *Rela*^{fl/+} cartilage (Fig. 7c). In heterozygous knockout chondrocytes, expression of the anti-apoptotic genes was decreased significantly, although the extent of the decrease in *Birc3* and *Pik3r1* expression was lower than that induced by homozygous knockout (Figs 5b and 7e).

To compare the transcriptional induction of target genes in homozygous and heterozygous knockout chondrocytes under the same conditions, we prepared primary chondrocytes from *Rela*^{fl/fl} and *Rela*^{fl/+} littermate neonates, and deleted *Rela* by adenoviral introduction of Cre recombinase (Fig. 8a). In Cre-overexpressing *Rela*^{fl/fl} and *Rela*^{fl/+} chondrocytes, the mRNA levels of *Traf2*, *Birc2*, *Birc3*, *Pik3r1*, *Adamts5* and *Hif2a* were decreased, and their mRNA levels in Cre-overexpressing *Rela*^{fl/+} chondrocytes were significantly higher than those in Cre-overexpressing *Rela*^{fl/fl} chondrocytes (Figs 5b, 7e and 8a). Next, we inhibited IKK using BMS-345541 instead of genetic deletion of *Rela* (Fig. 8b). Notably,

*Hif2a* expression was markedly decreased by 2.5 and 5 μM BMS-345541 treatment with or without TNF stimulation, while the anti-apoptotic genes were not significantly downregulated by 2.5 μM BMS-345541 treatment (Fig. 8b). Finally, we overexpressed Rela by an adenovirus in *Rela* knockout chondrocytes treated with 10 ng ml^{−1} TNF (Fig. 8c). Although the mRNA levels of all target genes were restored by exogenous Rela overexpression in a dose-dependent manner, *Traf2*, *Birc2* and *Birc3* were restored by the Rela adenovirus at 20 multiplicity of infection, whereas the restoration of *Pik3r1*, *Adamts5* and *Hif2a* required an multiplicity of infection of 100 (Fig. 8c).

## Discussion

The present study showed that Rela protects chondrocytes from apoptosis through induction of several anti-apoptotic genes. Loss-of-function of Rela resulted in mild impairment of skeletal growth and severe degeneration of articular cartilage. During skeletal development, homozygous knockout of *Rela* leads to

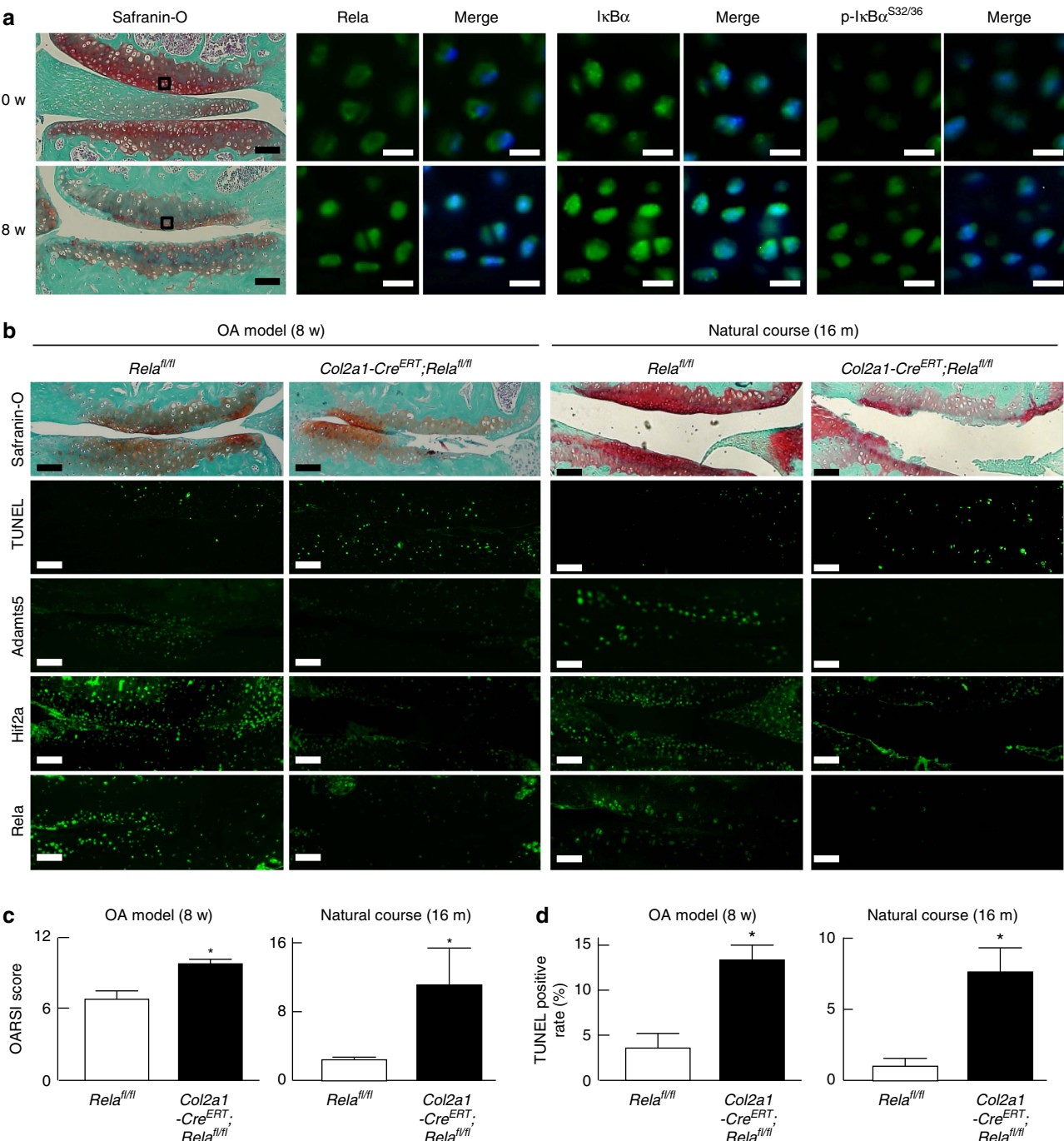

**Figure 4 | Regulation of OA development by Rela.** (**a**) Safranin-O staining and immunofluorescence of Rela, IκBα, and Ser 32 and 36 dual phosphorylated IκBα proteins in knee joint cartilage of an 8-week-old mouse (0 w) and at 8 weeks after surgery (8 week) to induce OA. Boxes in the Safranin-O staining panels indicate the regions of immunofluorescence. The images are representative of three independent experiments. Scale bars, 100 and 10 μm for Safranin-O staining and immunofluorescence, respectively. (**b**) Safranin-O, TUNEL staining and immunofluorescence of Adamts5, Hif2a and Rela in knee joint cartilage of $Rela^{fl/fl}$ and $Col2a1$-$Cre^{ERT}$; $Rela^{fl/fl}$ mice at 8 weeks after surgery (OA model, $n = 7$ and 10, respectively, left panels), and $Rela^{fl/fl}$ and $Col2a1$-$Cre^{ERT}$; $Rela^{fl/fl}$ mice at 16 months of age (natural course, $n = 5$ for each, right panels). Tamoxifen induction was performed at 7 weeks. The images are representative of three independent experiments. Scale bars, 100 μm. (**c**) OARSI (Osteoarthritis Research Society International) score of OA development. Data are expressed as the means ± s.d. of the mice per group. *$P < 0.05$ versus $Rela^{fl/fl}$ mice; Welch's t-test. (**d**) Rate of TUNEL-positive cells in articular cartilage. Data are expressed as the means ± s.d. of five mice per group. *$P < 0.05$ versus $Rela^{fl/fl}$ mice; Student's t-test.

impaired growth through ectopic apoptosis of chondrocytes (Fig. 2), whereas heterozygous knockout of *Rela* does not alter growth (Supplementary Fig. 5). In articular cartilage, homozygous knockout of *Rela* led to marked acceleration of OA through

enhanced apoptosis of chondrocytes (Fig. 4b,c), whereas heterozygous knockout of *Rela* resulted in suppression of OA development through inhibition of catabolic gene expression (Fig. 7a,b). These data indicate that Rela exerts anti-apoptotic and

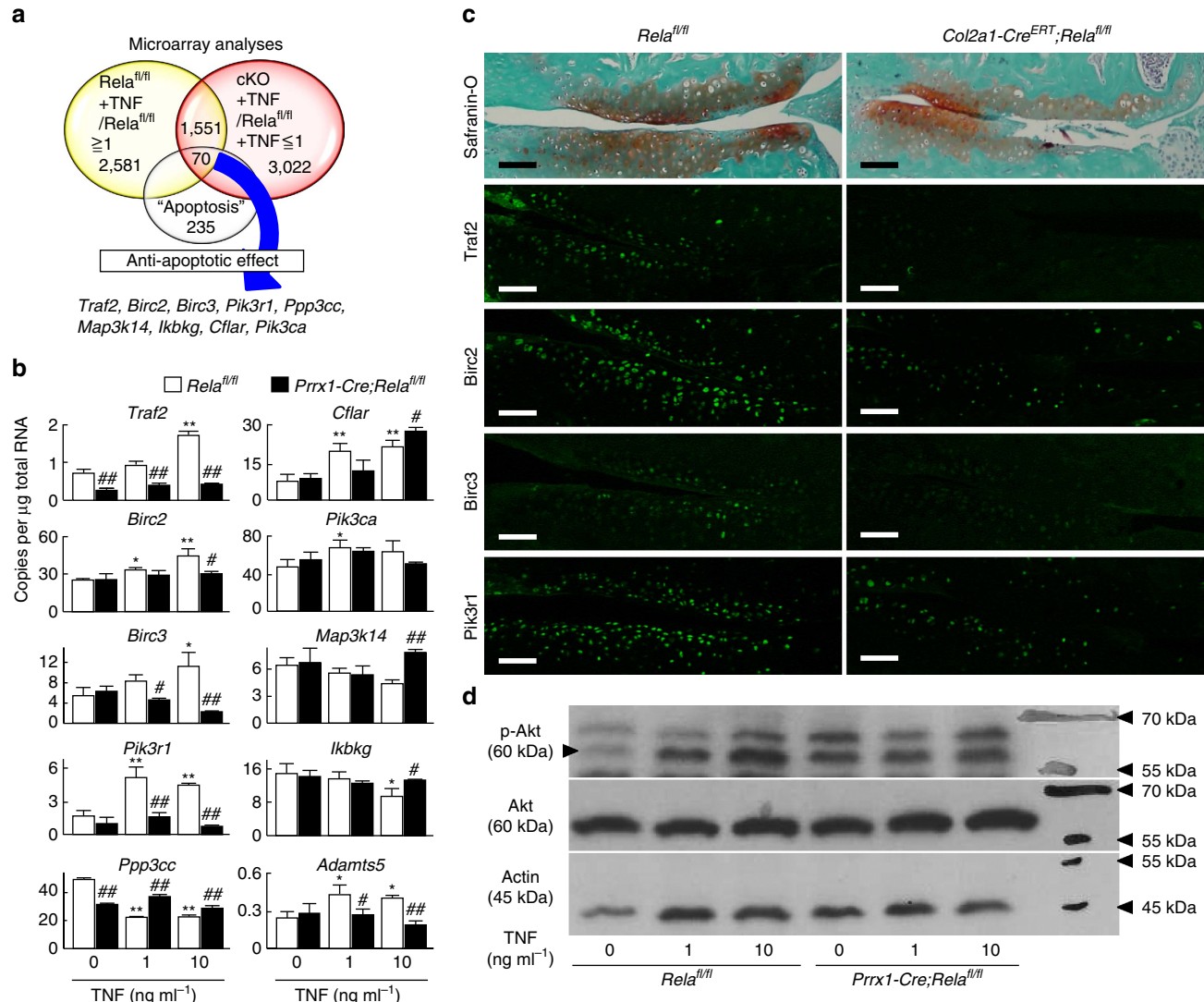

**Figure 5 | Transcriptional target genes of Rela.** (**a**) Schematic diagram to identify candidate genes by microarray analyses. cKO, *Prrx1-Cre; Rela*[fl/fl]. (**b**) mRNA levels of candidate genes and *Adamts5* in primary chondrocytes obtained from *Rela*[fl/fl] and *Prrx1-Cre; Rela*[fl/fl] (P5) mice with or without TNF treatment. Data are expressed as the means ± s.d. of three wells per group. #$P<0.05$, ##$P<0.01$ versus the respective *Rela*[fl/fl]control treated with the same dose of TNF; Student's *t*-test. *$P<0.05$, **$P<0.01$ versus *Rela*[fl/fl]cells without TNF treatment; Student's *t*-test. (**c**) Safranin-O staining and immunofluorescence of Traf2, Birc2, Birc3, Pik3r1 and Rela in knee joint cartilage of *Rela*[fl/fl] and *Col2a1-Cre*[ERT]; *Rela*[fl/fl] littermates at 8 weeks after surgery. The images are representative of three independent experiments. Scale bars, 100 μm. (**d**) Immunoblotting of total cell lysates of *Rela*[fl/fl] and *Prrx1-Cre; Rela*[fl/fl] chondrocytes using antibodies against Ser-473-phosphorylated Akt (p-Akt), Akt and actin with or without TNF treatment. The images are representative of two independent experiments.

catabolic effects in a biphasic manner, and that haploinsufficient expression of Rela can sustain chondrocyte survival and suppress articular cartilage degeneration.

In terms of chondrocyte differentiation during endochondral ossification, previous *in vitro* studies imply the involvement of Rela in various activities such as chondrogenesis, cell survival and cartilage matrix production[17–19,34]. However, it had been unknown how Rela regulates these steps *in vivo* because homozygous deletion of *Rela* allele leads to embryonic lethality at 15–16 days of gestation, which is concomitant with massive degeneration of the liver by apoptosis[35]. Tissue-specific knockout of *Rela* using *Rela-flox* mice revealed that Rela is indispensable for cell survival in the intestines and liver[36,37]. Similarly, the present data showed that chondrocyte-specific homozygous knockout of *Rela* slightly impaired skeletal growth, which was caused by enhanced apoptosis of chondrocytes. Furthermore, Rela was

revealed to be dispensable for the regulation of chondrogenesis and matrix production *in vivo*.

In the present study, homozygous knockout of *Rela* resulted in mild impairment of skeletal growth and severe degeneration of articular cartilage. *In vitro* experiments showed that the down-regulation of the anti-apoptotic genes by Rela deletion was further enhanced by TNF stimulation (Fig. 8). These results indicate that the anti-apoptotic effect of Rela in chondrocytes is more essential under stress loading. A recent study showed that the differentiation and turn-over of mouse articular chondrocytes occur over several months or more[38]. In mouse embryonic limb cartilage, chondrocytes differentiate in a much shorter period. These findings indicate that the survival time of a chondrocyte in adult articular cartilage is much longer than that in embryonic limb cartilage. Therefore, chondrocytes in adult articular cartilage may be exposed to various stresses and are more sensitive to decreased

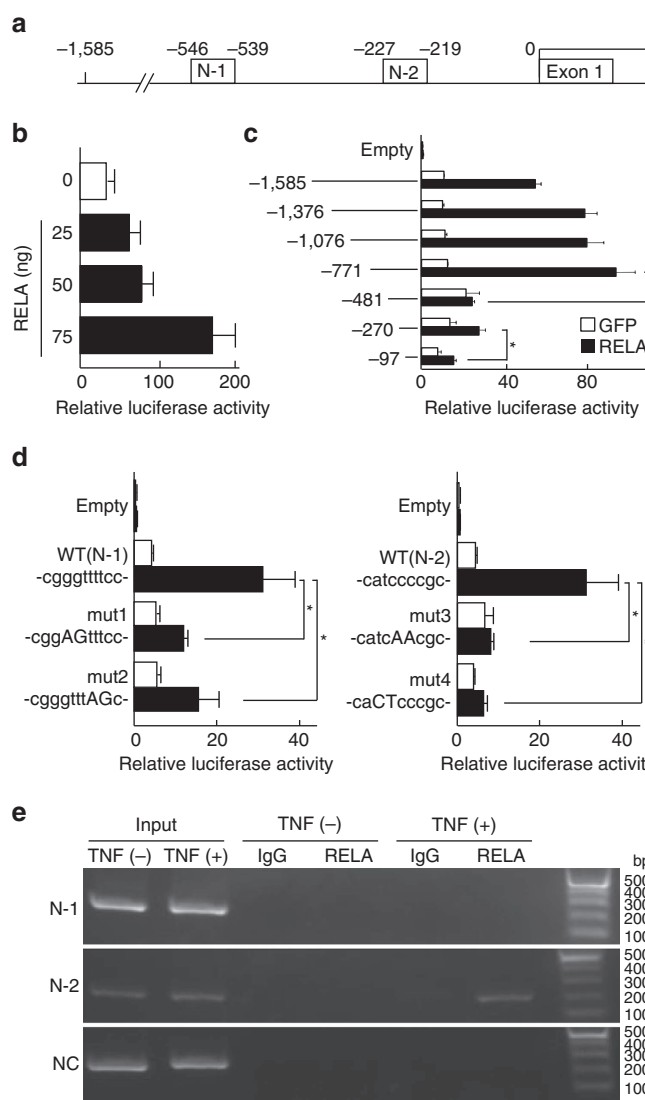

**Figure 6 | Transcriptional regulation of *PIK3R1* by RELA.** (**a**) A proximal promoter region of the human *PIK3R1* gene with two NF-κB consensus sequences, N-1 and N-2. (**b**) Luciferase activities resulting from RELA transfection of ATDC5 cells with a reporter construct containing a fragment of the *PIK3R1* promoter ( −1,585 to 0 bp relative to the transcription start site). Data are shown as means ± s.d. of three wells per group. (**c**) 5′-deletion analyses of the *PIK3R1* promoter. Data are shown as means ± s.d. of three wells per group. *P < 0.05, **P < 0.01; Student's *t*-test. (**d**) Site-directed mutagenesis analysis of the *PIK3R1* promoter fragment ( −1,585 to 0 bp). Mut1 and mut2 are mutations of N-1, and mut3 and mut4 are those of N-2. Data are shown as means ± s.d. of three wells per group. *P < 0.05; Student's *t*-test. (**e**) ChIP-PCR with lysates of OUMS-27 cells treated with or without TNF and amplified by a primer set spanning N-1 ( −593 to −408 bp), N-2 ( −285 to −77 bp) or not spanning both (NC, +55 to +289 bp) before (input) and after immunoprecipitation with the anti-RELA antibody or nonimmune IgG. The images are representative of two independent experiments.

expression of anti-apoptotic genes caused by Rela knockout compared with those in embryonic limb cartilage.

To delineate the status of the canonical NF-κB signalling pathway, we examined subcellular localization of Rela protein and phosphorylation of IκB protein by immunofluorescence. In embryonic limb cartilage, IkB phosphorylation and subcellular localization of Rela protein were not altered during chondrocyte

differentiation. Similar statuses of both proteins were observed in normal articular cartilage. These findings indicate that the canonical NF-κB signalling pathway is not strongly activated in developing limb cartilage or normal healthy articular cartilage. In contrast, IkB was phosphorylated and Rela was translocated into the nucleus in OA cartilage. According to previous studies, excessive mechanical stress induces various inflammatory mediators such as Adamts5 and TNF in chondrocytes[39,40]. In addition, mechanical loading can modulate the activity of the NF-κB signalling in a magnitude-dependent manner[41]. These data indicate that the canonical NF-κB signalling pathway may be activated by inflammatory mediators and excessive mechanical loading. Interestingly, homozygous gene deletion of *Rela* decreased the expression of both the anti-apoptotic and catabolic genes, but treatment with an IKK inhibitor at a low dose did not decrease the anti-apoptotic gene expression (Fig. 8a,b). Taken together, appropriate regulation of IKK or canonical NF-κB signalling activities may be a potent means to prevent OA development, although the physiological function of Rela should be protected to maintain articular cartilage.

We identified Pik3r1 as a novel transcriptional target gene of Rela (Figs 5 and 6). p85α, encoded by Pik3r1, is a regulatory subunit of PI3K and contributes to activation of Akt. Many previous studies have shown that PI3K/Akt signalling activates NF-κB signalling[33,42,43]. The PI3K/Akt-NF-κB signalling axis is one of the most promising targets for cancer and autoimmune disease therapies. The present data showed that upregulation of Pik3r1 by TNF stimulation was suppressed by Rela knockout or IKK inhibition (Figs 5b, 7e and 8a–c). Additionally, TNF stimulation did not upregulate phosphorylation of Akt in Rela knockout chondrocytes (Fig. 5d). Considering all these data, the PI3K/Akt-NF-κB signalling axis may induce *Pik3r1* expression in various tissues or cells other than chondrocytes and may act as a booster of signal transduction or form a kind of positive feedback loop. The transcriptional regulation of *Pik3r1* by Rela may be involved in oncogenesis or inflammatory disorders in addition to OA, because PI3K, Akt and NF-κB are associated with them.

In conclusion, we found that Rela exerts anti-apoptotic effects on chondrocytes during skeletal growth and articular cartilage homeostasis, but enhances cartilage catabolism upon activation of canonical NF-κB signalling. Our findings regarding the biphasic regulation of chondrocytes by Rela contribute to understanding the pathophysiology of OA and may provide a clue for novel treatments of OA.

## Methods

**Cell culture.** The mouse chondrogenic cell line ATDC5 (RIKEN Cell Bank, Tsukuba, Japan) was cultured in Dulbecco's modified Eagle's medium (DMEM)/ F12 (1:1) with 5% foetal bovine serum (FBS). The human chondrocyte cell line OUMS-27 (JCRB Cell Bank, Osaka, Japan) was cultured in DMEM with 10% FBS. The cell lines were not tested for mycoplasma contamination. Primary costal chondrocytes were isolated from the ribs of C57BL/6J neonates, and primary articular chondrocytes were isolated from 6-day-old C57BL/6J mice according to the standard protocol using collagenase D (ref. 44). Primary chondrocytes were cultured in DMEM with 10% FBS.

**Animals.** All experiments were performed according to protocols approved by the Animal Care and Use Committee of The University of Tokyo. In each experiment, we compared the genotypes of littermates maintained in a C57BL/6J background. *Col2a1-Cre* (ref. 22) and *Prrx1-Cre* mice[21] were purchased from the Jackson Laboratory. *Col2a1-Cre^{ERT}* mice[26] and *Rela^{fl/fl}* mice[24] were generously provided by Professor Fanxin Long (Washington University, St Louis) and Professor Roland M. Schmid (Technical University of Munich), respectively. *Col10a1-Cre* mice were generated by insertion of IRES-Cre-polyA cassette into the 3′-untranslated region[23].

To generate *Prrx1-Cre; Rela^{fl/fl}*, *Col2a1-Cre; Rela^{fl/fl}*, *Col10a1-Cre; Rela^{fl/fl}*, and *Col2a1-Cre^{ERT}; Rela^{fl/fl}* mice, *Rela^{fl/fl}* mice were mated with *Prrx1-Cre, Col2a1-Cre, Col10a1-Cre,* or *Col2a1-Cre^{ERT}* mice to obtain *Prrx1-Cre;Rela^{fl/+}*, *Col2a1-Cre;Rela^{fl/+}*, *Col10a1-Cre;Rela^{fl/+}* and *Col2a1-Cre^{ERT};Rela^{fl/+}* mice, respectively,

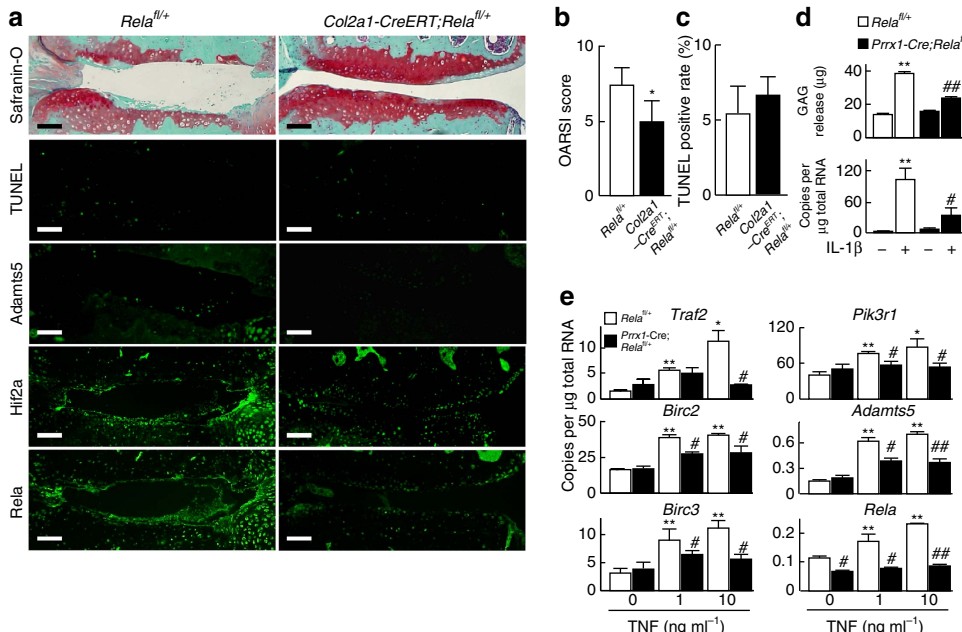

**Figure 7 | Suppression of OA development by heterozygous knockout of Rela.** (**a**) Safranin-O, TUNEL staining and immunofluorescence of Adamts5, Hif2a and Rela in knee joint cartilage of *Rela*<sup>fl/+</sup> and *Col2a1-Cre*<sup>ERT</sup>;*Rela*<sup>fl/+</sup> littermates at 8 weeks after surgery to induce OA. $n = 15$ and 14, respectively. Scale bars, 100 μm. (**b**) OARSI score of OA development. Data are expressed as the means ± s.d. of five mice per group. *$P < 0.05$ versus *Rela*<sup>fl/+</sup> mice; Welch's *t*-test. (**c**) Rate of TUNEL-positive cells in articular cartilage. Data are expressed as the means ± s.d. of five mice per group. (**d**) Amount of glycosaminoglycans released into the medium determined by the dimethylmethylene blue assay during 4 days of culture of femoral heads obtained from *Rela*<sup>fl/+</sup> and *Prrx1-Cre; Rela*<sup>fl/+</sup> littermates ($n = 4$ for each) with or without interleukin-1β treatment (top), and the mRNA level of *Adamts5* in the femoral heads (bottom). Data are expressed as the means ± s.d. of three wells per group. **$P < 0.01$ versus *Rela*<sup>fl/+</sup> mice without interleukin-1β treatment; Student's *t*-test. #$P < 0.05$, ##$P < 0.01$ versus *Rela*<sup>fl/+</sup> mice with interleukin-1β treatment; Student's *t*-test. (**e**) mRNA levels of the target genes in primary chondrocytes obtained from *Rela*<sup>fl/+</sup> and *Prrx1-Cre; Rela*<sup>fl/+</sup> mice (P5) with or without TNF treatment. Data are expressed as the means ± s.d. of three wells per group. *$P < 0.05$, **$P < 0.01$ versus *Rela*<sup>fl/+</sup> mice without TNF treatment; Student's *t*-test. #$P < 0.05$, ##$P < 0.01$ versus the respective *Rela*<sup>fl/+</sup> control treated with the same dose of TNF; Student's *t*-test.

and then mated with Rela<sup>fl/fl</sup> mice. Sequences of the primers used for genotyping are shown in Supplementary Table 1.

**Histology.** Haematoxylin and eosin (H&E) staining was performed according to standard protocols after fixation in 4% paraformaldehyde/PBS. For immunohistochemistry, the sections were incubated with antibodies against Rela (1:100; #8242, Cell Signaling Technology, Danvers, MA), IκBα (1:100; sc-371, Santa Cruz Biotechnology, Santa Cruz, CA), p-IκBα<sup>S32/36</sup> (1:100; sc-101713, Santa Cruz Biotechnology), Col2a1 (1:500; LB-1297, LSL, Tokyo, Japan), Col10a1 (1:500; LB-0097, LSL), Adamts5 (1:500; sc-134952, Santa Cruz Biotechnology), Hif2a (1:100; sc-28706, Santa Cruz Biotechnology), Traf2 (1:500; ab126578, Abcam, Cambridge, UK), Birc2 (1:500; ab25939, Abcam), Birc3 (1:500; sc7944, Santa Cruz Biotechnology) and Pik3r1 (1:500; sc423, Santa Cruz Biotechnology) diluted in blocking reagent. For immunofluorescence, we used a CSA II Biotin-Free Catalyzed Amplification System (Agilent Technologies, Santa Clara, CA) and applied Hoechst 33258 (Agilent Technologies) for counterstaining. Double staining of mouse embryo skeletons was performed with a solution containing Alizarin red S and Alcian blue 8GX (Sigma, St Louis, MO) after fixation in 99.5% ethanol and acetone. For BrdU labelling, BrdU (Sigma) was injected intraperitoneally into pregnant mice before killing, and the sections were stained using a BrdU Immunohistochemistry System (Merck Millipore, Darmstadt, Germany). For TUNEL staining, we used an *in situ* Apoptosis Detection Kit (Takara Bio, Otsu, Japan). Immunohistochemistry was performed at least three times for each analysis. For radiological analysis, plain radiographs were obtained using a soft X-ray apparatus Softex CMB-2 (Softex, Ebina, Japan).

**Cell proliferation assay.** Primary chondrocytes were seeded at $1 \times 10^3$ cells per well in a 96-well plate. The cell proliferation was examined using an XTT Assay Kit (Roche, Mannheim, Germany) at the indicated time points. The absorbance of the reaction product was quantified using a MTP-300 microplate reader (Corona Electric, Hitachinaka, Japan).

**Real-time RT-PCR.** Total RNA from cells was isolated with an RNeasy Mini Kit (Qiagen, Hilden, Germany) according to the manufacturer's instructions. An aliquot (1 μg) was reverse transcribed with QuantiTect Reverse Transcription

(Qiagen) to prepare single-stranded cDNA. Real-time RT-PCR was performed with an ABI Prism 7000 Sequence Detection System (Applied Biosystems, Foster City, CA) using QuantiTect SYBR Green PCR Master Mix (Qiagen) according to the manufacturer's instructions. Standard plasmids were synthesized with a ZERO Blunt II TOPO Cloning Kit (Invitrogen, Carlsbad, CA) according to the manufacturer's instructions. All reactions were run in triplicate. Copy numbers of target gene mRNAs in each total RNA were calculated by reference to standard curves and adjusted to mouse standard total RNA (Applied Biosystems) with rodent glyceraldehyde-3-phosphate dehydrogenase as an internal control. The primer sequences are shown in Supplementary Table 2.

**Western blotting.** Cells were lysed in M-PER Mammalian Protein Extraction Reagent (Thermo Scientific, Waltham, MA). The lysates were fractionated by sodium dodecyl sulphate-polyacrylamide gel electrophoresis and transferred onto nitrocellulose membranes (BIO-RAD, Hercules, CA). After blocking with 6% dry skim milk, the membranes were incubated with primary antibodies against Akt (1:500; #4685, Cell Signaling Technology), p-Akt (1:500; #4060, Cell Signaling Technology) and actin (1:10,000; A2066, Sigma). The membranes were then incubated with a horseradish peroxidase-conjugated antibody (Promega, Madison, WI). Immunoreactive proteins were visualized with ECL Plus (GE Healthcare Lifescience, Chicago, IL). Original images of the immunoblots were shown in Supplementary Fig. 6.

**OA experiments.** Tamoxifen (Sigma; 100 μg per g of body weight) was intraperitoneally injected into 7-week-old Col2a1-Cre<sup>ERT</sup>; Rela<sup>fl/fl</sup>, Col2a1-Cre<sup>ERT</sup>; Rela<sup>fl/+</sup> mice and their respective control mice daily for 5 days. We then performed the surgical procedure to establish an experimental OA model in 8-week-old male mice[25]. Under general anaesthesia, resection of the medial collateral ligament and the medial meniscus was performed using a surgical microscope. The mice were analysed at 8 weeks after surgery. We also analysed OA development with aging using 1-year-old mice bred under physiological conditions. After the surgery, all mice were maintained under the same conditions (three mice per cage). OA severity was quantified by the OARSI system[45], which was assessed by a single observer who was blinded to the experimental groups.

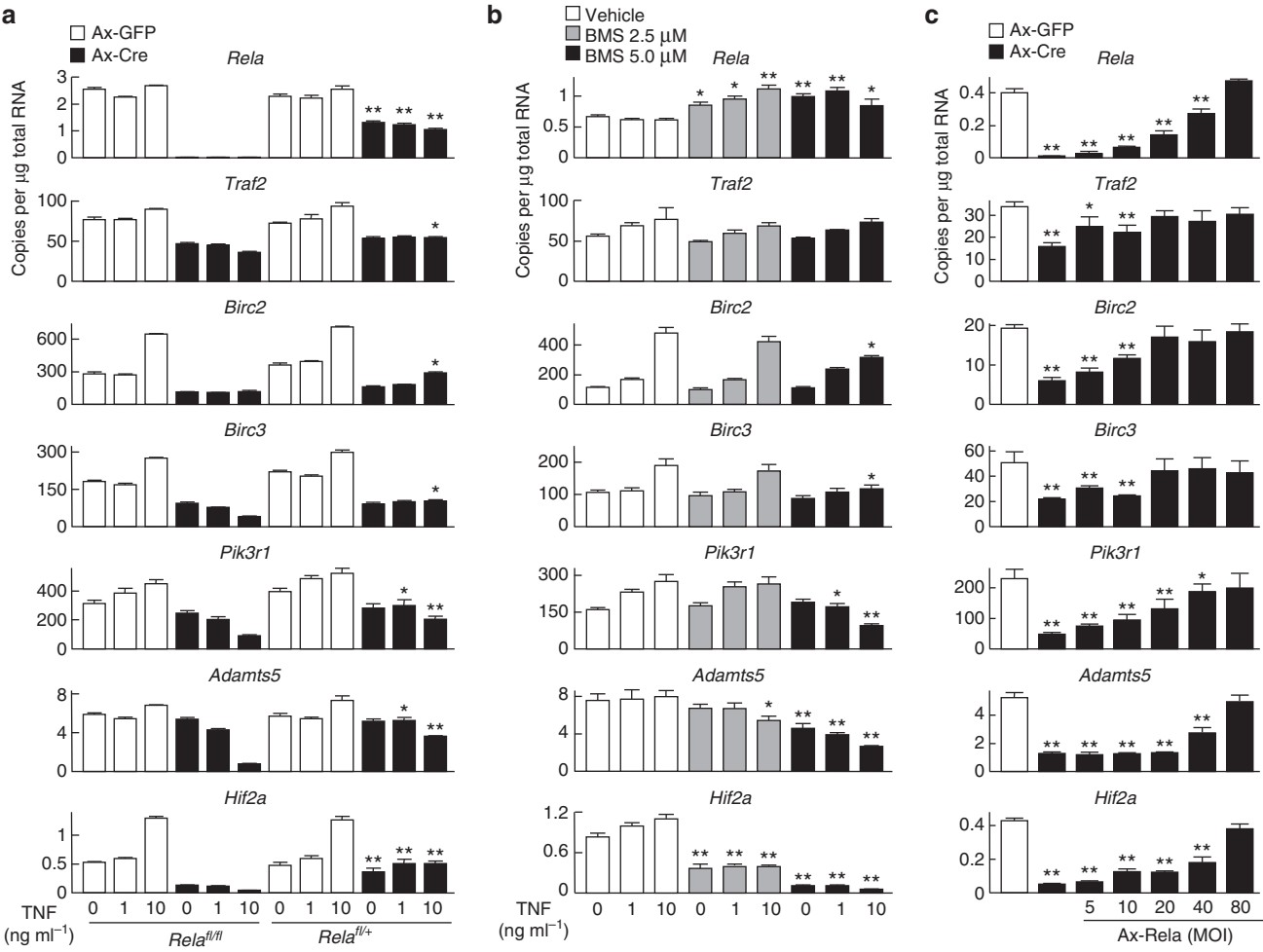

**Figure 8 | Transcription of the target genes under various conditions of Rela or NF-κB signalling. (a)** mRNA levels of the target genes in *Rela^{fl/fl}* and *Rela^{fl/+}* primary chondrocytes overexpressing GFP (Ax-GFP) or Cre (Ax-Cre) by adenoviral induction with or without TNF treatment. Adenovirus infection was performed at a multiplicity of infection (MOI) of 50. Data are expressed as the means ± s.d. of three wells per group. *P < 0.05, **P < 0.01 versus Cre-overexpressing *Rela^{fl/fl}* chondrocytes treated with the same dose of TNF; Student's t-test. **(b)** mRNA levels of the target genes in *Rela^{fl/fl}* primary chondrocytes treated with DMSO (vehicle) or 2.5 and 5 μM BMS-345541 (IKK inhibitor) with or without TNF stimulation. Data are expressed as the means ± s.d. of three wells per group. *P < 0.05, **P < 0.01 versus the vehicle control treated with the same dose of TNF; Student's t-test. **(c)** mRNA levels of the target genes in *Rela^{fl/fl}* primary chondrocytes transduced with Cre and Rela by adenoviral vectors. GFP or Cre was transduced at an multiplicity of infection (MOI) of 20. The amount of Rela adenovirus varied from an MOI of 5–80. Data are expressed as the means ± s.d. of three wells per group. *P < 0.05, **P < 0.01 versus Ax-GFP; Student's t-test.

**Microarray analyses.** Primary chondrocytes were obtained from 6-day-old *Prrx1-Cre; Rela^{fl/fl}* and *Rela^{fl/fl}* littermates and treated with or without TNF (10 ng ml⁻¹) for 3 days. Total RNA was isolated from the cells with an RNeasy Mini Kit (Qiagen). Microarray experiments were performed using SurePrint G3 Mouse Gene Expression 8 × 60K (Agilent Technologies). The microarray data have been deposited in the Gene Expression Omnibus (www.ncbi.nlm.nih.gov/geo/) under accession no. GSE79233.

**Construction of expression vectors.** The coding sequence of RELA was amplified by PCR, and cloned into pCMV-HA vector[18]. Adenoviral vectors for GFP, Cre and Rela were generated by the AdenoX Expression System (Clontech, Palo Alto, CA)[19,46]. All vectors were verified by DNA sequencing.

**Luciferase assays.** We prepared the *PIK3R1* promoter region (from − 1585 to + 76 bp relative to the transcription start site) by PCR using human genomic DNA as the template, and cloned it into the pGL4.10[luc2] vector (Promega). Deletion and mutation constructs were prepared by PCR. Transfection of ATDC5 cells was performed in triplicate in 48-well plates using FuGENE 6 transfection reagent (Roche): FuGENE 6 with 150 ng plasmid DNA, 100 ng pGL4 reporter vector, 50 ng effector vector and 4 ng pRL-TK vector (Promega) for the internal control per well. For the dose-response experiment of RELA, we used 25, 50 and 75 ng of the expression vector. Luciferase assays were performed with a PicaGene Dual SeaPansy Luminescence Kit (Toyo Ink, Tokyo, Japan) using a GloMax 96

Microplate Luminometer (Promega). Data are presented as the ratio of firefly to Renilla activities.

**ChIP assay.** The ChIP assay was performed with a OneDay ChIP kit (Diagenode, Liege, Belgium) according to the manufacturer's instructions. *In vivo* crosslinking was performed after 1 day with or without TNF (10 ng ml⁻¹) treatment of OUMS-27 cells. Cell lysates were sonicated to shear the genomic DNA. For immuno-precipitation, we used an antibody against RELA (Cell Signaling Technology) and normal control rabbit IgG. We employed primer sets that amplified the area including N-1 (−593 to −408 bp), N-2 (−285 to −77 bp) or that not including the motif (+ 55 to + 289 bp). PCR was performed using KOD FX Neo (TOYOBO, Osaka, Japan). The primer sequences are shown in Supplementary Table 3. Original images of the ChIP-PCR were shown in Supplementary Fig. 7.

**Proteoglycan release assay.** Proteoglycan release was assessed as reported previously[47]. Briefly, femoral heads were harvested from 3-week-old *Rela^{fl/+}* and *Prrx1-Cre;Rela^{fl/+}* mice. The explants were cultured for 3 days with or without interleukin-1β (10 ng ml⁻¹) in DMEM. The proteoglycan content in the medium was measured as sulphated glycosaminoglycan by a colorimetric assay using dimethylmethylene blue.

**Statistical analyses.** Statistical analyses of experimental data were performed with the unpaired two-tailed Student's t-test when homogenous variances were assumed

($P > 0.05$) by the F-test, and with Welch's *t*-test when homogenous variances were not assumed ($P < 0.05$) by the F-test. *P*-values of $<0.05$ were considered significant.

**Data availability.** Microarray data that support the findings of this study have been deposited Gene Expression Omnibus with primary accession code GSE79233. All other relevant data are available from the authors.

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

## Acknowledgements

We thank Prof F. Long and Prof R.M. Schmid for generously providing *Col2a1-Cre*^ERT mice and *Rela*^fl/fl mice, and J. Sugita, H. Kawahara and K. Kaneko for technical assistance. This study was supported by grants-in-aid for Scientific Research from the Japanese Ministry of Education, Culture, Sports, Science and Technology (15K10460, 25253087 and 23689065). The sponsor had no role in the study design, data collection, data analysis, data interpretation or writing of the manuscript.

## Author contributions

H.Ko. and T.S. designed the study; H.Ko., S.H.C., D.M., S.I., M.H., Y.H., Y.T., K.O. and T.S. performed the experiments; H.Ko., Y.M., F.Y., U-i.C., H.Ak., H.Ka., S.T. and T.S. analysed the data; H.Ko. and T.S. wrote the manuscript.
