## [Peer Review File · Nature Communications]

Reviewers' comments:

Reviewer #1 (Remarks to the Author):

GENERAL COMMENTS: The studies reported in this manuscript constitute a detailed analysis of the regulation of apoptosis by RelA in mice comparing deficiency at different developmental stages and in adult cartilage and in relevant culture models. The title hints at the authors' interpretation of the results and conclusions, which are not very well stated in the Abstract, where the data summary is somewhat confusing with regard to differential effects of homozygous versus heterozygous RelA deficiency. Several points of clarification need to be addressed as listed below.

SPECIFIC COMMENTS:

1. Abstract:

- a. It is unclear what is meant that RelA/p65 is "representative factor". It is more precisely a key subunit that mediates canonical NF- κ B signaling.
- b. Please indicate at what age the tamoxifen-inducible homozygous knockout was performed that resulted in severe OA.
- c. Line 37: Please clarify what is being compared in "induced by a smaller amount of RelA compared with the catabolic genes".
- d. It is a little difficult to follow how the results differ in homozygous versus heterozygous conditional and inducible knockout of RelA in postnatal chondrocytes. This is because there seem to be differential regulation of apoptosis compared to proteinases. Is it true that the authors are attributing the cartilage degradation to apoptosis rather than to differential expression of proteinases?
- e. The summary of the data is easier to follow in the first paragraph of the Discussion. A similar systematic description could be used here in the Abstract. The hypothesis that "RelA may regulate articular cartilage homeostasis in a biphasic manner" stated on Page 16 in the Results does not seem to be recapitulated here or elsewhere.
- f. The conclusion that "IKK and NF- κ B signaling may represent potent therapeutic targets of OA" is not very specific to the outcome of the experiments reported here.

2. Introduction, pages 3-4: The description of proteins involved in NF- κ B signaling needs some clarification, as it does not indicate that the family members listed, including RelA, are subunits that act as transcription factors usually after heterodimerization; the inhibitors, I κ B, do not interact with all subunits, and that different IKKs complex together to mediate canonical (mainly IKK β) versus non-canonical (IKK α) NF- κ B signaling.

3. Results:

- a. Pages 7-8, Figure 3: It is not clear what is meant by "ectopic", since the apoptosis is affected by endogenous RelA deficiency.
- b. Figure 4: Mice were subjected to DMM surgery to induce OA at 8 weeks of age. This seems young compared to 10 to 12 weeks of age used by most laboratories.
- c. Page 18, Figure 8C: It is also not clear what is meant by "a smaller amount" of RelA required for upregulation of some genes compared to others. Since these graphs represent dose responses to adv-RelA, then it should be possible to explain this in terms of threshold and maximum doses (MOI).

4. Discussion:

- a. Page 20, lines 286-291: The emphasis of the explanation about the effects of TNF stimulation needs to be clarified. It seems that what you are trying to say is that the effects of RelA deficiency on anti-apoptotic gene expression cannot be observed in basal conditions in the absence of TNF stimulation because the cells need to be stressed.
- b. Pages 21-22: The authors make important points about the physiological role of RelA and potential biphasic effects in regulating apoptosis that need to be highlighted in the Abstract and elsewhere.

Reviewer #2 (Remarks to the Author):

Anti-apoptotic effects in chondrocytes during skeletal growth and biphasic regulation of articular cartilage by Rela

Overview: In this study the authors analyzed the in vivo functions of Rela in embryonic limbs and adult articular cartilage. Limb mesenchyme- or chondrocyte-specific homozygous knockout of Rela impaired skeletal development, but only slightly. In contrast, tamoxifen-inducible homozygous knockout resulted in severe osteoarthritis (OA) accompanied by enhanced chondrocyte apoptosis.

Specific comments: This is a nicely written and presented original paper. The major claims of the paper are important and novel. This paper should be of interest to others in the field of cartilage biology and osteoarthritis research. The work convincing, and no further evidence is required to strengthen the conclusions. The paper may potentially influence new research directions and thinking in the field.

Suggested improvements: The main issue is the title. The authors should attempt to develop a better title for this paper. This reviewer is not convinced that the current title actually works.

Reviewers' comments:

Reviewer #1 (Remarks to the Author):

GENERAL COMMENTS: The studies reported in this manuscript constitute a detailed analysis of the regulation of apoptosis by Rela in mice comparing deficiency at different developmental stages and in adult cartilage and in relevant culture models. The title hints at the authors' interpretation of the results and conclusions, which are not very well stated in the Abstract, where the data summary is somewhat confusing with regard to differential effects of homozygous versus heterozygous Rela deficiency. Several points of clarification need to be addressed as listed below.

SPECIFIC COMMENTS:

1. Abstract:

a. It is unclear what is meant that Rela/p65 is "representative factor". It is more precisely a key subunit that mediates canonical NF- κ B signaling.

A1. As the reviewer has suggested, we have changed the description.

b. Please indicate at what age the tamoxifen-inducible homozygous knockout was performed that resulted in severe OA.

A2. We have indicated the age in the Abstract.

c. Line 37: Please clarify what is being compared in "induced by a smaller amount of Rela compared with the catabolic genes".

A3. This phrase was deleted when we revised the Abstract. We have clarified this point in the last paragraph of the Results.

d. It is a little difficult to follow how the results differ in homozygous versus heterozygous conditional and inducible knockout of Rela in postnatal chondrocytes. This is because there seem to be differential regulation of apoptosis compared to proteinases. Is it true that the authors are attributing the cartilage degradation to apoptosis rather than to differential expression of proteinases?

A4. Proteinases were down-regulated in homozygous and heterozygous knockout mice. In addition, apoptosis was only enhanced in homozygous knockout mice. Similar findings were obtained in our *in vitro* experiments.

e. The summary of the data is easier to follow in the first paragraph of the Discussion. A similar systematic description could be used here in the Abstract. The hypothesis that "Rela may regulate articular cartilage homeostasis in a biphasic manner" stated on Page 16 in the Results does not seem to be recapitulated here or elsewhere.

A5. We have changed the Abstract and Discussion according to the reviewer's suggestion.

f. The conclusion that "IKK and NF-kB signaling may represent potent therapeutic targets of OA" is not very specific to the outcome of the experiments reported here.

A6. We have changed the concluding sentence in the Abstract.

2. Introduction, pages 3-4: The description of proteins involved in NF-kB signaling needs some clarification, as it does not indicate that the family members listed, including RelA, are subunits that act as transcription factors usually after heterodimerization; the inhibitors, Ikb, do not interact with all subunits, and that different IKKs complex together to mediate canonical (mainly IKKbeta) versus non-canonical (IKKalpha) NF-kB signaling.

A7. We have added the description in the third paragraph.

3. Results:

a. Pages 7-8, Figure 3: It is not clear what is meant by "ectopic", since the apoptosis is affected by endogenous RelA deficiency.

A8. We intended to state that chondrocyte apoptosis was enhanced by RelA deficiency. We have changed "ectopic" to "enhanced".

b. Figure 4: Mice were subjected to DMM surgery to induce OA at 8 weeks of age. This seems young compared to 10 to 12 weeks of age used by most laboratories.

A9. We usually employ the "medial model" by resection of the medial collateral ligament and medial meniscus (*Osteoarthritis Cartilage*, 13:632, 2005), and not the "DMM model". We established the medial model at 8 weeks of age, because skeletal development of mice is almost complete at this time. In our previous reports, we performed the medial model surgery at 8 weeks of age (*Nat Med.* 16:678-86, 2010, *Arthritis Rheum.* 64:198-203, 2012, *Hum Mol Genet.* 21:1111-23, 2012, *Ann Rheum Dis.* 72:748-53, 2013, *Proc Natl Acad Sci U S A.* 110:1875-80, 2013, *Ann Rheum Dis.* 73:2062-4, 2014, *Proc Natl Acad Sci U S A.* 112:3080-5, 2015, *Osteoarthritis Cartilage.* 24:688-97, 2016).

c. Page 18, Figure 8C: It is also not clear what is meant by "a smaller amount" of RelA required for upregulation of some genes compared to others. Since these graphs represent dose responses to adv-Rela, then it should be possible to explain this in terms of threshold and maximum doses (MOI).

A10. We agree with the reviewer. As the reviewer suggested, we have changed the description.

4. Discussion:

a. Page 20, lines 286-291: The emphasis of the explanation about the effects of TNF stimulation needs to be clarified. It seems that what you are trying to say is that the effects of RelA deficiency on anti-apoptotic gene expression cannot be observed in basal conditions in the absence of TNF stimulation because the cells need to be stressed.

A11. We have changed the description in accordance with the present data.

b. Pages 21-22: The authors make important points about the physiological role of Rela and potential biphasic effects in regulating apoptosis that need to be highlighted in the Abstract and elsewhere.

A12. We have changed the Abstract according to the reviewer's suggestion.

Reviewer #2 (Remarks to the Author):

Anti-apoptotic effects in chondrocytes during skeletal growth and biphasic regulation of articular cartilage by Rela

Overview: In this study the authors analyzed the *in vivo* functions of Rela in embryonic limbs and adult articular cartilage. Limb mesenchyme- or chondrocyte-specific homozygous knockout of Rela impaired skeletal development, but only slightly. In contrast, tamoxifen-inducible homozygous knockout resulted in severe osteoarthritis (OA) accompanied by enhanced chondrocyte apoptosis.

Specific comments: This is a nicely written and presented original paper. The major claims of the paper are important and novel. This paper should be of interest to others in the field of cartilage biology and osteoarthritis research. The work convincing, and no further evidence is required to strengthen the conclusions. The paper may potentially influence new research directions and thinking in the field.

Suggested improvements: The main issue is the title. The authors should attempt to develop a better title for this paper. This reviewer is not convinced that the current title actually works.

A1. We have changed the title to "biphasic regulation of chondrocytes by Rela through induction of anti-apoptotic and catabolic target genes".

REVIEWERS' COMMENTS:

Reviewer #1 (Remarks to the Author):

The authors have done a nice job of responding to my comments on the original manuscript. I have no further comment.